# Attenuation of Smooth Muscle Cell Phenotypic Switching by Angiotensin 1-7 Protects against Thoracic Aortic Aneurysm

**DOI:** 10.3390/ijms232415566

**Published:** 2022-12-08

**Authors:** Anshul S. Jadli, Noura N. Ballasy, Karina P. Gomes, Cameron D. A. Mackay, Megan Meechem, Tishani Methsala Wijesuriya, Darrell Belke, Jennifer Thompson, Paul W. M. Fedak, Vaibhav B. Patel

**Affiliations:** 1Department of Physiology and Pharmacology, Cumming School of Medicine, University of Calgary, Calgary, AB T2N 4N1, Canada; 2Libin Cardiovascular Institute, University of Calgary, Calgary, AB T2N 4N1, Canada; 3Section of Cardiac Surgery, Department of Cardiac Sciences, Cumming School of Medicine, University of Calgary, Calgary, AB T2N 4N1, Canada; 4Alberta Children’s Hospital Research Institute, Cumming School of Medicine, University of Calgary, Calgary, AB T2N 4N1, Canada

**Keywords:** thoracic aortic aneurysm (TAA), angiotensin II, angiotensin 1-7, vascular smooth muscle cells, phenotypic switch

## Abstract

Thoracic aortic aneurysm (TAA) involves extracellular matrix (ECM) remodeling of the aortic wall, leading to reduced biomechanical support with risk of aortic dissection and rupture. Activation of the renin-angiotensin system, and resultant angiotensin (Ang) II synthesis, is critically involved in the onset and progression of TAA. The current study investigated the effects of angiotensin (Ang) 1-7 on a murine model of TAA. Male 8–10-week-old ApoEKO mice were infused with Ang II (1.44 mg/kg/day) and treated with Ang 1-7 (0.576 mg/kg/day). ApoEKO mice developed advanced TAA in response to four weeks of Ang II infusion. Echocardiographic and histological analyses demonstrated increased aortic dilatation, excessive structural remodelling, perivascular fibrosis, and inflammation in the thoracic aorta. Ang 1-7 infusion led to attenuation of pathological phenotypic alterations associated with Ang II-induced TAA. Smooth muscle cells (SMCs) isolated from adult murine thoracic aorta exhibited excessive mitochondrial fission, oxidative stress, and hyperproliferation in response to Ang II. Treatment with Ang 1-7 resulted in inhibition of mitochondrial fragmentation, ROS generation, and hyperproliferation. Gene expression profiling used for characterization of the contractile and synthetic phenotypes of thoracic aortic SMCs revealed preservation of the contractile phenotype with Ang 1-7 treatment. In conclusion, Ang 1-7 prevented Ang II-induced vascular remodeling and the development of TAA. Enhancing Ang 1-7 actions may provide a novel therapeutic strategy to prevent or delay the progression of TAA.

## 1. Introduction

Aortic aneurysm (AA) is characterized by matrix degeneration of the aortic wall resulting in permanent focal dilatation of the aorta to more than 50% of the normal diameter. Thoracic AA (TAA) occurs at an estimated frequency of 5–10 persons per 100,000 per year, with a death rate of ∼70% among untreated patients [1,2]. Although onset and progression of TAA is associated with excessive extracellular matrix (ECM) remodeling, inflammation, and phenotypic switching of vascular smooth muscle cells (VSMCs) [3,4], precise molecular mechanisms contributing to its pathogenesis remain elusive. Inadequate understanding of the underlying mechanisms involved in the onset and progression of TAA has critically impacted the development of novel therapeutic targets.

The renin-angiotensin system (RAS) has been implicated in the development of thoracic and abdominal aneurysms [5]. Angiotensin II (Ang II), an effector octapeptide of RAS, has been shown to promote the development of aneurysms by inducing hemodynamic alterations (hypertension), oxidative stress, inflammation, and vascular remodeling in the aorta [6,7]. Previous studies have reported TAA induced by infusion of Ang II in mice with several distinct aortic pathologies such as aortic wall thickening, elastin fragmentation, progressive luminal dilatation, and macrophage accumulation [8,9]. As the Ang II-induced TAA model closely approximates human disease, studies validated the use of this murine model to understand human aortic aneurysm pathogenesis.

To antagonize the detrimental effects of Ang II in aneurysms, angiotensin receptor blockers and angiotensin-converting enzyme (ACE) inhibitors have been tested in clinical trials [10]. Beta-blockers, doxycycline (a broad-spectrum matrix metalloproteinase (MMP) inhibitor), anti-inflammatory agents (cyclooxygenase inhibitors), immunosuppressants (rapamycin), and HMG Co-A reductase inhibitors were tested in preclinical and clinical investigations [11,12,13]. None of these pharmacological approaches have proven to be effective in limiting the progression and/or risk of rupture in aortic aneurysms. Currently, there is no clinically approved therapy for TAA; prophylactic surgical replacement of the dilated aorta is only recommended when the inherent risk of aortic surgery is lower than the risk of a serious clinical event, including catastrophic aortic rupture [14]. These surgical interventions include repair (insertion of an intraluminal graft via open access to the aneurysmal aorta) or endovascular stenting [15]. Due to potential operative risk and the inability to treat the early stages of the aneurysms, these surgical procedures remain effective only in preventing aortic rupture. The absence of clinically relevant therapeutic interventions against TAA is the consequence of the elusive pathophysiology of initiation and progression of this deadly disease. 

Angiotensin 1-7 (Ang 1-7), a heptapeptide, produced by ACE2-mediated proteolytic cleavage of Ang II, was shown to antagonize the adverse effects of Ang II in various vascular diseases, including hypertension and atherosclerosis [16,17]. ACE2 and Ang 1-7 exert protective effects in atherosclerosis via protection of endothelial cell function and inhibition of Ang II-induced inflammatory response [18]. Activation of the Ang 1-7/Mas receptor (MasR) pathway attenuated vascular remodeling and inflammation resulting in mitigation of intracranial aneurysm in a preclinical murine model [19]. A recent study demonstrated the anti-apoptotic and anti-inflammatory effects of Ang 1-7 attenuating the abdominal aortic aneurysm formation in a murine model [20]. The therapeutic effects of Ang 1-7 in TAA remain to be investigated. In the present study, we sought to determine the effects of Ang 1-7 on vascular remodeling and TAA. 

## 2. Results

### 2.1. Ang 1-7 Prevents Ang II-Induced TAA

Angiotensin infusion to ApoEKO mice has been shown to induce hemodynamic changes along with dyslipidemia, resulting in adverse vascular remodeling and development of aortic aneurysm [21]. As previously shown [21], four weeks of Ang II-infusion to ApoEKO mice led to the development of TAA (Figure 1). Aortic dilatation, a defining characteristic of TAA, was measured using transthoracic echocardiography. Representative echocardiography images and their quantification validated the onset of aortic dilatation in response to chronic Ang II infusion (Figure 1A–D). Given its potent vasculo-protective properties, including mitigation of adverse vascular remodeling [22], the effects of chronic Ang 1-7 administration on TAA were assessed. Chronic intraperitoneal infusion of Ang 1-7 led to attenuation of pathological changes associated with Ang II-induced TAA, evident by the significantly reduced thoracic aortic diameters in the Ang 1-7-treatment group compared to the Ang II group (Figure 1B). Although Ang 1-7 administration mitigated the Ang II-induced aortic dilatation (Figure 1C,D), the aortic expansion index, an indicator of biomechanical elasticity and recoil property of the aorta, was not significantly altered in the thoracic aorta of Ang II (±Ang 1-7)-infused mice (Figure 1E). Aortic wall distensibility provides a reliable, sensitive, and quantifiable measurement of aortic wall properties and is commonly used to characterize aortic stiffness and risk of aortic rupture [23]. Aortic wall distensibility is described as the ability of the aorta to expand during systole. The aortic distensibility in the thoracic aorta was measured using the high-frequency ECG-gated Kilohertz Visualization (EKV)-mode of echocardiography. Thoracic aorta displayed markedly reduced aortic wall distensibility in response to Ang II-infusion, which was prevented with Ang 1-7 co-treatment (Figure 1F). Echocardiographic measurements of aortic dilatation and biomechanical properties uncovered the loss of structural and functional integrity of the thoracic aorta in Ang II-infused mice which were prevented by Ang 1-7 (Figure 1).

### 2.2. Ang 1-7 Mitigates Adverse Vascular Remodeling in a Murine Model of TAA

To examine the impact of chronic administration of Ang II and Ang 1-7 on aortic wall vascular remodeling, principal ECM proteins associated with the structural integrity of the aortic wall were assessed. Histological analysis using Verhoeff-Van Gieson (VVG) staining displayed elastin fibers with significantly reduced thickness in the thoracic aorta of Ang II-infused mice when compared to saline controls (Figure 2A). In contrast, chronic Ang 1-7-infusion preserved the structural integrity of elastin fibers (Figure 2A). In addition to disorganized elastin fibers, Ang II-infusion also induced significant medial degeneration resulting in decreased thoracic aortic media thickness, a clinical feature commonly observed in patients with aortic aneurysms. Chronic Ang 1-7 administration mitigated the Ang II effects on the medial degeneration, resulting in preserved medial thickness (Figure 2B). 

ECM remodeling is involved in the onset and progression of TAA [24,25]. Histological evaluation of collagen, a major constituent of the extracellular matrix, by Gomori trichrome staining revealed excessive deposition of collagen in the adventitia of Ang II-infused aorta, leading to marked perivascular fibrosis (Figure 2C). Ang II-induced perivascular fibrosis was also corroborated by Picro-Sirius red staining, confocal imaging, and quantification of collagen content (Figure 2D,E). Chronic Ang 1-7 administration significantly reduced maladaptive aortic structural remodeling and perivascular fibrosis resulting in decreased collagen levels and their deposition in the peri-adventitial region (Figure 2B–E). 

Immunofluorescence staining and confocal image analysis showed severely decreased alpha-smooth muscle actin (α-SMA) levels in the medial smooth muscle cells (Figure 2F,G). Being a major contractile protein that forms the core of sarcomeres in SMCs, decreased α-SMA levels indicate VSMC “phenotypic switch” from a contractile to a synthetic phenotype in the Ang II-infusion group. Chronic administration of Ang 1-7 restored the levels of α-SMA, which suggests a crucial role of Ang 1-7 in reinforcing the contractile phenotype of VSMCs (Figure 2F,G). Our data also suggest that the protective effects of Ang 1-7 on TAA may be, at least partly, mediated via the effects of Ang 1-7 on VSMCs.

### 2.3. Ang 1-7 Treatment Decreases Perivascular Adipose Tissue Inflammation by Reducing Macrophage Maturation and Polarization

Histological analyses using hematoxylin and eosin staining showed increased inflammatory cell infiltration in the peri-aortic adipose tissue (Figure 3A). Perivascular adipose tissue (PVAT) surrounding the thoracic aorta displays brown adipose tissue-like characteristics [26], with the adipose progenitor cells of thoracic aortic PVAT having the ability to differentiate into uncoupling protein-1 (UCP-1)-positive adipocytes. Ang II-induced TAA was associated with increased adipocyte hypertrophy (Figure 3B), a well-characterized feature of adipose tissue dysfunction and whitening of adipocytes [27,28]. Adipose tissue remodeling in Ang II-induced TAA was further validated by increased mRNA expression of *Mmp2*, *Mmp9*, and *Tnfa* (Figure 3C–E). Importantly, chronic Ang 1-7 administration attenuated adipocyte hypertrophy and inflammatory cell infiltration, suggesting mitigation of Ang II-induced PVAT remodeling. Although, mRNA expression for *Mmp2*, *Mmp9*, and *Tnfa* were not statistically significant, Ang 1-7-mediated alleviation of adipose tissue remodeling also resulted in decreased mRNA expressions of *Mmp2*, *Mmp9*, and *Tnfa* (Figure 3C–E). Immunofluorescence staining and confocal imaging of the PVAT displayed markedly increased levels of pan macrophage marker F4/80^+^ in response to Ang II administration (Figure 3F,G). Notably, Ang II administration also promoted their polarization to CD163^+^ macrophages, a phenotype previously shown to be strongly associated with fibrotic response (Figure 3H). Notably, chronic administration of Ang 1-7 reduced levels of F4/80^+^, CD163^+^, and F4/80^+^CD163^+^ macrophages in the PVAT, resulting in reduced adipose tissue inflammation and fibrotic remodeling (Figure 3G–I).

### 2.4. Ang II-Induced Phenotypic Switching and Hyperproliferation of Thoracic Aortic SMCs Are Attenuated by Ang 1-7

As the outcome of in vivo experiments suggested the potential effects of Ang 1-7 in reinforcing the contractile phenotype in VSMCs (Figure 2F,G), we sought to carry out mechanistic investigations using isolated thoracic aortic VSMCs. Ang II has previously been shown to induce phenotypic switching in aortic VSMCs, i.e., the polarization of contractile VSMCs to synthetic and/or osteo/chondrogenic phenotype [29]. VSMC phenotypic switch has been consistently observed to contribute to the development and progression of various vascular diseases, including aortic aneurysms. Synthetic VSMCs promote ECM remodeling and vascular inflammation, playing a pivotal role in the growth of an aortic aneurysm [30,31,32]. We assessed the effect of Ang 1-7 treatment on phenotypic switching of VSMCs derived from the murine thoracic aorta. The quantitative mRNA expression analysis revealed significantly reduced expression of contractile genes, including *Acta2*, *Cnn1*, and *Myh11* (Figure 4A–C) in Ang II-treated thoracic VSMCs. Moreover, Ang II treatment also increased the mRNA expressions of *Il-6, Mmp2, Mmp9, Col1a1,* and *Col3a1* (Figure 4D–H) in the VSMCs, signifying their polarization towards inflammatory and synthetic phenotypes. Notably, co-treatment with Ang 1-7 preserved the contractile phenotype of thoracic aortic SMCs, as evident by preserved mRNA expressions of *Acta2*, *Cnn1*, and *Myh11* (Figure 4A–C). Ang 1-7 treatment also mitigated Ang II-induced onset of inflammatory and synthetic phenotype in thoracic aortic SMCs, resulting in reduced mRNA expressions of *Il-6*, *Mmp2*, *Mmp9*, *Col1a1*, and *Col3a1*. Though Ang 1-7 treatment attenuated Ang II-induced elevated *Il-6* expression, the higher mRNA expression of *Il-6* in the Ang 1-7 group compared to controls suggested a milder effect of Ang 1-7 on inflammatory cytokines than other parameters. (Figure 4D–H). Ang II-treated dedifferentiated VSMCs exhibit increased cell proliferation [33]. Ki67, a nuclear protein associated with cell proliferation, was used as a marker of hyperproliferation. Ang II treatment significantly increased cell proliferation in thoracic aortic SMCs, as indicated by increased Ki67 fluorescence intensity in flow cytometric analysis (Figure 4I,J). Ang 1-7 attenuated Ang II-mediated hyperproliferation in thoracic aortic SMCs (Figure 4I,J). In vitro experiments using murine thoracic aortic SMCs corroborated the critical role of SMC phenotypic change in the Ang II-induced vascular remodeling and aortic aneurysm. Attenuation of SMC phenotypic change by Ang 1-7 plays a key role in its vasoprotective, anti-fibrotic, and anti-inflammatory effects. Thus, gene expression data and attenuation of Ang II-induced macrophage infiltration in the thoracic aorta suggested anti-inflammatory effects of Ang 1-7 in TAA.

### 2.5. Ang 1-7 Attenuates Ang II-Mediated Mitochondrial Fission and ROS Generation in Thoracic Aortic SMCs

In vivo and in vitro experiments suggest a key role of Ang 1-7 in reinforcing the contractile phenotype of VSMCs and protecting against the Ang II-induced development of TAA. Recently, a study investigating the effects of mitochondrial fission on Ang II-induced hypertension demonstrated the role of mitochondrial dynamics in VSMC phenotypic switching [34]. To investigate the effect of Ang II on mitochondria, we used MitoTracker Red CMXRos staining and confocal imaging. Mitotracker Red staining of thoracic VSMCs showed significantly increased mitochondrial fission in response to treatment with Ang II (Figure 5A,B). Quantification of Mitotracker Red-stained confocal images using MiNA, an NIH ImageJ plugin, showed decreased mean branch length, mean network branches, and mitochondrial footprint, indicative of smaller mitochondria in Ang II-treated thoracic aortic SMCs (Figure 5C–E). Treatment of thoracic aortic SMCs with Ang 1-7 administration exhibited elongated mitochondria along with preserved mean branch length, mean network branches, and mitochondrial footprint, suggesting alleviation of Ang II-induced mitochondrial fission (Figure 5C–E). 

Increased cellular and mitochondrial ROS generation are associated with dynamic mitochondrial changes, smooth muscle phenotypic changes, and the progression of TAA [34,35,36]. We sought to investigate the effect of Ang 1-7 on cellular and mitochondrial ROS generation. The superoxide indicator dihydroethidium (DHE) which produces fluorescence upon oxidation by superoxide in live cells was used for the evaluation of cellular ROS. Confocal imaging of DHE stained-thoracic aortic SMCs and quantification of DHE fluorescence intensity showed significantly increased cellular ROS levels in Ang II-treated VSMCs (Figure 5F,G). Staining of thoracic aortic SMCs with MitoSOX Red mitochondrial superoxide indicator and flow cytometric analysis revealed increased mitochondrial ROS generation in Ang II-treated VSMCs (Figure 5H,I). Ang 1-7 treatment significantly attenuated cellular and mitochondrial ROS levels, as evident in the representative confocal images and flow cytometric analysis of DHE-stained and MitoSOX-stained VSMCs, respectively (Figure 5F–I). Ang 1-7 prevents the development and progression of TAA via inhibition of mitochondrial dynamics, ROS generation, SMC phenotypic switch, and PVAT inflammation.

## 3. Discussion

An aortic aneurysm is a complex multifactorial cardiovascular disease affecting the population worldwide [37]. Metabolic, genetic, and environmental factors contribute to pathological progression and, sometimes, rupture of the aorta. Despite being the 13th leading cause of death in North America, most cases of aortic aneurysms remain undetected until severe clinical events, including rupture, which is associated with a 90% mortality rate from ruptured aneurysms [37,38]. Treatment alternatives such as β-blockers, tetracyclines, statins, and ACE inhibitors/angiotensin receptor blockers have been proposed to reduce inflammation and progression of aortic dilatation [38]. However, evidence-based support for the effectiveness of such non-validated pharmacologic interventions is weak. Open surgical repair and endovascular stents remain the mainstay of treatments for aneurysm management. Targeted pharmacological therapies based on underlying cellular and molecular mechanisms are lacking, but could be clinically significant to prevent or delay the progression of aortic dilatation and the growing need for prophylactic aortic surgery. 

Ang 1-7, a vasoactive heptapeptide, has shown potent vasculoprotective effects against hypertension, atherosclerosis, intracranial aneurysm, abdominal aortic aneurysm, and heart failure in various experimental investigations [18,39,40,41]. Xue et al. recently identified the ability of Ang 1-7 to alleviate the severity of Ang II-induced abdominal aortic aneurysm [20]. Aortic aneurysms are known to be heterogeneous diseases, with distinct mechanisms involved in the onset of thoracic and abdominal aneurysms. Multisegmental aneurysms are observed in less than 10% of the patients with aortic aneurysms [37]. Moreover, region-specificity of aortic aneurysms varies in differential prevalence, risk factors, and responses to treatment. Differences between thoracic and abdominal aortic aneurysms are thought to be due to structural heterogeneity between the thoracic and abdominal aorta, leading to different pathophysiological mechanisms involved in the onset and progression of thoracic and abdominal aortic aneurysms [42,43]. The therapeutic potential of Ang 1-7 in TAA remains to be investigated. 

To investigate the effects of Ang 1-7 in TAA, we employed a murine model combining two of the major risk factors of TAA, i.e., hyperlipidemia and activation of the renin-angiotensin system leading to hemodynamic abnormalities. ApoEKO mice developed severe and reproducible TAA in response to chronic Ang II administration. Being one of the most widely used murine models, Ang II infusion in ApoEKO mice has been characterized to result in ECM remodeling and perivascular fibrosis, inflammation, luminal expansion, and aortic dissection [7,8,21]. Numerous studies have identified Ang 1-7 effects that antagonize Ang II-induced detrimental defects in cardiovascular pathologies [17,18,39,40,41,44,45,46]. In the present study, we identified the protective effect of Ang 1-7 against the development of TAA. 

TAA is characterized by pathological remodeling of the aortic wall leading to excessive ECM remodeling, specifically medial elastolysis and perivascular collagen deposition [25]. Ang 1-7 is a potent anti-fibrotic peptide via its actions, resulting in decreased activation of myofibroblasts and decreased activation of MMP2 and MMP9 [47]. Consistently, chronic administration of Ang 1-7 resulted in preserved elastin fibers along with reduced perivascular collagen levels. In addition to previously known effects of Ang 1-7 on fibroblasts, our data suggest a key contribution of thoracic aortic SMC polarization in the anti-fibrotic and anti-elastolytic effects of Ang 1-7 in TAA. VSMCs are central in maintaining vascular tone in the vessel wall [48]. However, owing to their remarkable plasticity, VSMCs are well-known to undergo a phenotypic switch to synthetic, senescent, adipocytic, osteochondrogenic, and foam cell-like phenotypes [49]. Emerging evidence links contractile to synthetic phenotypic switch in aortic SMCs with vascular remodeling and aneurysm formation [30]. The synthetic phenotype of VSMCs is characterized by reduced expression of contractile proteins and increased production and secretion of MMPs, inflammatory cytokines, and collagen [48,49]. The synthetic phenotype of VSMCs promotes vascular inflammation and ECM remodeling, augmenting vascular pathology [50]. The VSMC phenotypic switching has been identified to be a key mechanistic observation in various aortopathies, including Marfan syndrome, Ehlers–Danlos syndrome, Loey–Dietz syndrome, and Turner syndrome [51,52,53,54]. Phenotypic switching of thoracic aortic VSMCs has been identified as a predisposing factor in the development of TAA [31,32]. Our results strongly suggest that Ang II-induced VSMC polarization to synthetic phenotype plays a key role in the development of TAA. Importantly, by reinforcing the contractile phenotype of VSMCs, Ang 1-7 alleviated ECM remodeling and prevented the development of TAA. Although our results highlight the critical role of VSMCs, aortic adventitial fibroblasts (AoAF) are also implicated in perivascular fibrosis, ECM remodeling, and aneurysm formation [55]. Effects of Ang 1-7 on AoAF and its relative contribution to anti-fibrotic and anti-elastolytic effects of Ang 1-7 remain to be investigated. 

A recent discovery determined the critical involvement of mitochondrial dynamics and mitochondrial ROS in Ang II-induced VSMC phenotypic switching [34]. Inhibition of mitochondrial fission by mitochondrial division inhibitor-1 (mdivi-1) restored mitochondrial structure and activity and attenuated Ang II-induced phenotypic switching in VSMCs [34]. The study emphasized the role of Drp1 phosphorylation in increased mitochondrial fission leading to Ang II-stimulated VSMC phenotypic switching. Similarly, Lu et al. demonstrated Drp1-dependent Ang II-induced VSMC phenotypic switching and attenuation by diallyl trisulfide [56]. The dedifferentiation of VSMCs has been characterized by increased proliferation and migration [57]. VSMC proliferation has been observed as a pathological response in TAA, along with disrupted TGF-β signaling [58]. Recently, Pei et al. demonstrated a significantly increased VSMC proliferation in ascending aortic aneurysm, which induced elastin and collagen synthesis via the OPN/p38 MAPK signaling pathway [59]. Previous studies reported cell proliferation as a major hallmark of phenotypic switching [57]. The causal effect of ROS generation on VSMCs proliferation via distinct signalling pathways have been well documented [60]. The PKC-dependent activation of ERK1/2 has been implicated in O_2_-mediated VSMCs proliferation [61]. Alternatively, cyclophilin A, a secreted chaperone protein induced by oxidative stress, promotes H_2_O_2_-dependent VSMCs proliferation [62,63]. The upregulation of NOX subunits and increased ROS generation in cardiovascular pathologies involving vascular remodelling indicated the association of oxidative stress with vascular remodelling [64,65]. Accumulating evidence suggested an association of the generation of ROS with aberrant mitochondrial morphology [66]. Conversely, genetic ablation of pro-mitofusion genes or inducible deletion of Drp1 leads to altered mitochondrial morphology and generation of ROS [67,68,69]. Mechanistically, we speculate a reciprocal association between mitochondrial dynamics and ROS generation during Ang II-induced VSMCs phenotypic switching in TAA. In the current study, we found that Ang II treatment led to increased mitochondrial fission and oxidative stress in thoracic aortic VSMCs. The altered mitochondrial dynamics and increased ROS generation were associated with the synthetic phenotype of VSMCs, evident by increased inflammatory and matrix remodeling markers, reduced contractile markers, and VSMC proliferation. Ang 1-7 attenuated Ang II-induced mitochondrial fission and subsequently preserved the synthetic phenotype of thoracic aortic VSMCs (Figure 6). 

Although the PVAT surrounding the conduit arteries was considered mechanical support, a growing body of evidence suggests its pathological association with cardiovascular diseases [70]. A recent study suggests that local inflammation of PVAT contributes to the development of metabolic syndrome [71]. Loss of ACE2 has been associated with epicardial adipose tissue (EAT) inflammation and macrophage activation in response to a high-fat diet. Ang 1-7 treatment ameliorated the lipotoxicity, EAT inflammation, and polarization of macrophages to a pro-inflammatory phenotype [72]. The activation of the Ang 1-7/Mas axis leads to reduced expression of proinflammatory cytokines. The PVAT Ang II type 1a receptor (AT1aR) has been implicated in vascular inflammation and aneurysm development [71]. Previously, deletion of AT1aR has been shown to result in markedly attenuated aortic aneurysm development, macrophage infiltration, and proteolytic activity in the abdominal aorta [71]. Our study exhibited that Ang 1-7 altered *Mmp2* expression in the PVAT. Although no significant impact was observed on *Mmp9* and *Tnfa* mRNA expressions, promising, and perhaps selective, impact on *Mmp2* expression warrants further investigation. Moreover, future studies investigating the impact of Ang 1-7 on PVAT phenotypic change may provide novel mechanistic insights into crosstalk between PVAT and the aortic SMCs. In congruence with previous reports, our study identified Ang II-induced PVAT inflammation, associated with increased levels of F4/80^+^, CD163^+^, and F4/80^+^CD163^+^ macrophages in the PVAT surrounding the thoracic aorta. Increased levels of activated macrophages may be due to either increased infiltration or maturation of resident adipose tissue monocytes. CD163^+^ macrophages are known to be critically involved in fibrotic responses in various tissues and may play a key role in increased perivascular fibrosis in Ang II-induced TAA [73]. The Ang 1-7-mediated reduction in CD163^+^ macrophages indicated reduced adverse fibrotic remodeling in the perivascular tissue. The multifactorial vasculoprotective effects of Ang 1-7 treatment can be attributed to the amelioration of PVAT inflammation, activation, the polarization of macrophages, and adverse perivascular fibrosis. In addition to effects on VSMC phenotypic change, anti-fibrotic effects of Ang 1-7 may also be mediated by a reduction in macrophage polarization to a fibrotic phenotype. 

Using a preclinical model of TAA, our study provided strong scientific evidence for the beneficial effects of Ang 1-7 against pathological changes predisposing to the development and progression of TAA. The suppression of Ang II-induced pathological phenotypic switching, inflammation, oxidative stress, and excessive mitochondrial fission, and thus attenuation of TAA progression, suggest the therapeutic potential of Ang 1-7 in TAA. Further clinical studies will be required for the corroboration of promising preclinical results. Based on the outcome of the clinical trial, appropriate dosage and the route of Ang 1-7 administration can be recommended.

## 4. Materials and Methods

### 4.1. Experimental Animals

*ApoE*-deficient mice (ApoEKO; C57BL/6 background) were purchased from the Jackson Laboratory (Bar Harbor, Me). Mice were housed in pathogen-free conditions and had access to sterilized food and water ad libitium. All the animal experiments were performed at the department of Physiology and Pharmacology of Cumming School of Medicine at the University of Calgary. Alzet micro-osmotic pumps (Model 1004, Durect Corp., Cupertino, CA, USA) were implanted subcutaneously in 8–10-week-old male mice to deliver Ang II (1.44 mg/kg/day) or saline (control) for 4 weeks [22]. A subgroup of ApoEKO mice receiving Ang II was also implanted with osmotic pumps to deliver Ang 1-7 (0.576 mg/kg/day) intraperitoneally. The study was approved by the Institutional Ethics Committee of the University of Calgary (AC17-0055 & AC21-0029). All experiments were conducted as per the guidelines of the University of Calgary Animal Care and Use Committee and the Canadian Council of Animal Care.

### 4.2. Echocardiography

Ultrasonic images of the aorta were obtained using a Vevo 3100 high-resolution imaging system equipped with a real-time micro visualization scan head (MX550D, VisualSonics) in mice anesthetized with 2% isoflurane mixed with O_2_, as previously described [22,74]. The diameters of the thoracic aorta were measured using M-mode images of echocardiography. The maximum aortic lumen diameter (systolic aortic diameter corresponding to cardiac systole) and the minimum aortic lumen diameter (diastolic aortic diameter corresponding to cardiac diastole) monitored by simultaneous ECG recordings were measured and used to calculate the aortic expansion index [(Systolic aortic diameter—Diastolic aortic diameter)/Systolic diameter X 100]. The aortic distensibility was analyzed using B-mode as well as EKV images of the thoracic aorta by VEVO Vasc software (version 5.7.0, VisualSonics, Toronto, ON, Canada).

### 4.3. Isolation and Culture of Vascular Smooth Muscle Cells

VSMCs were isolated from thoracic aortas harvested from 10–12-week-old ApoEKO mice and were cultured as previously described [22,75]. Briefly, aortas were harvested in aseptic conditions, and periaortic adipose tissues were promptly removed. Thoracic aortas were incubated with a digestive enzyme solution containing 1 mg/mL collagenase II (#LS004174, Worthington Biochemical, Lakewood, NJ, USA) and 1 mg/mL elastase (#LS002292, Worthington Biochemical, Lakewood, NJ, USA) at 37 °C in a 5% CO_2_ incubator for 10 min. The adventitial and inner layers of the aorta were then removed using a surgical microscope. Medial layers of the aortas were cut into small pieces and incubated with the digestive enzyme solution for 90 min at 37 °C in a 5% CO_2_ incubator. After digestion, cells were washed and cultured in a sterile DMEM/F12 cell culture medium (#11330057, Gibco, Waltham, MA, USA) containing 20% fetal bovine serum (FBS, #26140079, Gibco) supplemented with 1% penicillin/streptomycin (#CA12001-712, VWR, Mississauga, ON, Canada). Cells were passaged upon reaching 90–100% confluency. Thoracic aortic SMCs were serum-deprived for 24 h by incubation in DMEM/F12 medium with 1% fetal bovine serum and penicillin/streptomycin. Cells were subsequently challenged with Ang II (100 nM; #05-23-0101, Millipore Sigma, St. Louis, MO, USA) in DMEM/F12 for 24 h. In the treatment group, Ang 1-7 (100 nM) (#H-1715.0025, Bachem, Bubendorf, BL, Switzerland) was added to the VSMCs 30 min before Ang II exposure. A subgroup of VSMCs without Ang II and Ang 1-7 treatments served as controls.

### 4.4. Histological Analysis and Immunofluorescence Staining

Mice were perfusion-fixed at 80 mmHg with buffered-formalin after 4 weeks of Ang II (±Ang 1-7) or saline infusion, allowing the blood vessels to be fixed in their native state, as previously described [22]. Subsequently, aortas were collected and fixed in 10% buffered formalin for 48 h. Then, 5 μm thick formalin-fixed paraffin-embedded (FFPE) sections were used for the histological staining, including hematoxylin and eosin (H&E), Gomori trichrome, and Verhoeff Van Gieson (VVG), as previously described [22]. Picro-Sirius red (PSR) staining was carried out as previously described [16]. The collagen positive area in the thoracic aorta was imaged using a confocal microscope (Leica SP8) and quantified by the morphometric analysis using the Fiji/ImageJ (version 1.52i, Bethesda, MD, USA) software. Then, 5 μm thick formalin-fixed paraffin-embedded (FFPE) sections were used for the immunofluorescence staining as previously described [22]. Briefly, after deparaffinization, sections were immersed in EDTA-citrate buffer (pH 6.2) for antigen retrieval. Thereafter, sections were washed with PBS and permeabilized with Triton X-100 (0.1%), followed by blocking with bovine serum albumin (BSA; 1.5%). Sections were incubated with corresponding primary antibodies for α-smooth muscle actin (α-SMA, 1:500, #ab32575, Abcam, Trumpington, CB, UK), F4/80 (1:100, #sc-52664, Santa Cruz Biotechnology, Dallas, TX, USA), or CD163 (1:100, #ab182422, Abcam, Trumpington, CB, UK), overnight at 4 °C, followed by secondary antibodies for Alexa Fluor 488-conjugated Donkey anti-Rat (1:1000, #A21208, Invitrogen, Waltham, MA, USA), Alexa Fluor 555-conjugated Goat anti-Rabbit (1:1000, #A32732, Invitrogen, Waltham, MA, USA), or Alexa Fluor 647-conjugated Goat anti-Mouse (1:1000, #A32728, Invitrogen, Waltham, MA, USA) at room temperature for 1 h. ProLong Gold with DAPI (#P36935, Invitrogen, Waltham, MA, USA) was used as a mounting medium and nuclear staining. Immunostained aorta sections were imaged using a line-scanning confocal microscope (Leica SP8) and analyzed using the Fiji/ImageJ (version 1.52i, Bethesda, MD, USA) software.

### 4.5. Mitochondrial Structural Analysis

The mitochondrial structure was assessed using MitoTracker Red CMXRos (#M7512, Invitrogen, Waltham, MA, USA) staining and confocal imaging. Briefly, control or Ang II (±Ang 1-7)-treated cells were incubated with 100 nM Mitotracker Red CMXRos for 30 min at 37 °C in a CO_2_ incubator. Mitotracker Red-labelled cells were fixed with 4% paraformaldehyde and imaged using the Leica SP8 confocal microscope. Mitochondrial fragmentation was quantified using Fiji/ImageJ (version 1.52i, Bethesda, MD, USA) and the Mitochondrial Network Analysis (MiNA) plugin [76]. 

### 4.6. Cellular Reactive Oxygen Species (ROS) Analysis

Cellular ROS levels were analyzed using the dihydroethidium (DHE) staining as previously described [22]. Briefly, control and Ang II (±Ang 1-7)-treated cells were incubated with DHE (10 μM; #D11347, Invitrogen) at 37 °C for 30 min in a CO_2_ incubator. The cells were immediately imaged using a confocal microscope (Leica SP8). Quantitative measurement of DHE fluorescence intensity was carried out using the Fiji/ImageJ (version 1.52i, Bethesda, MD, USA) software.

### 4.7. Mitochondrial ROS Analysis

Mitochondrial ROS (mROS) generation was assessed using MitoSOX Red mitochondrial superoxide indicator (#M36008, Invitrogen). Control and Ang II (±Ang 1-7)-treated thoracic VSMCs were trypsinized, and monolayers, as well as floating cells in the cell culture medium, were centrifuged at 1500× *g* for 5 min. Cell pellets were washed and resuspended in HBSS/Ca^2+^ buffer at 0.5 × 10^6^ cells/mL. Cells were then incubated with 1 µM MitoSOX Red at 37 °C for 10 min in the dark. After the incubation, cells were washed and resuspended in HBSS/Ca^2+^ buffer for flow cytometric analysis using the Attune NxT flow cytometer (ThermoFisher Scientific, Waltham, MA, USA) at excitation and emission wavelengths of 510/580 nm. The Invitrogen™ Attune™ NxT Software (version 2.6, ThermoFisher Scientific, Waltham, MA, USA) was used to analyze the MitoSOX fluorescence intensity, representing the mitochondrial ROS levels.

### 4.8. Assessment of VSMC Proliferation

Thoracic aortic SMC proliferation was evaluated using the Ki67 staining and flow cytometric analysis. Briefly, VSMCs were plated at 25,000 cells per well in 6 well plates and treated with saline and Ang II (±Ang 1-7) for 24 h. Control and Ang II (±Ang 1-7)-treated cells were trypsinized, and monolayer cells in the cell culture medium were centrifuged at 1500g rpm for 5 min. The cell pellets were fixed with 70% ethanol and incubated at −20 °C for 2 h. After fixation, cells were washed twice with staining buffer (1% FBS, 0.09% NaN_3_ in PBS) and resuspended in staining buffer at 1 × 10^6^ cells/mL. The cells were then incubated with the Anti-Ki67 antibody (#ab16667, Abcam, Trumpington, CB, UK) at room temperature for 30 min. After the incubation, cells were washed twice with staining buffer. Alexa Fluor 488-conjugated Goat anti-Rabbit secondary antibody (#A11034, Invitrogen, Waltham, MA, USA) was added, and cells were incubated at room temperature for 30 min in the dark. After incubation, cells were washed twice with staining buffer, and samples were analyzed within 1 h using an Attune NxT flow cytometer (ThermoFisher Scientific, Waltham, MA, USA) at 488 nm excitation and 530 nm emission wavelength. Quantitative analysis of flow cytometric data was performed using Invitrogen™ Attune™ NxT Software (version 2.6, ThermoFisher Scientific, Waltham, MA, USA). 

### 4.9. Quantitative Real-Time PCR

Total RNA was extracted from thoracic aortic SMCs and PVAT using the TRIzol reagent (#15-596-018, Thermo Fisher Scientific, Waltham, MA, USA) according to the manufacturer’s instructions. The extracted RNA was dissolved in RNase-free water, and the concentration of the total RNA was determined with a spectrophotometer. Next, cDNA was prepared using the cDNA Reverse Transcription Kit (#4374966, Thermo Fisher Scientific, Waltham, MA, USA). Following this, TaqMan probes for quantitative PCR were used to measure the expression of specified genes: *Acta2* (Mm.PT.58.16320644), *Cnn1* (Mm.PT.58.12652862), *Myh11* (Mm.PT.58.9236105), *Il-6* (Mm.PT.58.10005566), *Mmp2* (Mm00439498_m1), *Mmp9* (Mm00442991_m1), *Col1a1* (Mm00801666_g1), *Col3a1* (Mm00802331_m1), *Tnfa* (Mm.PT.58.12575861), and *18s rRNA* (Hs.PT.39a.22214856.g). For each gene, a standard curve was generated using known concentrations of cDNA (0.1, 1, 10, 100, and 1000 ng) as a function of cycle threshold (CT). Expression analysis of the reported genes was performed by TaqMan Real-time PCR using the QuantStudio™5 system (Thermo Fisher Scientific, Waltham, MA, USA). Data were analyzed using QuantStudio™ design and analysis software (version 1.4.3, Thermo Fisher Scientific, Waltham, MA, USA). All samples were analyzed in triplicates in 384 well plates. The mRNA expression data were presented as relative expressions to 18S rRNA, an endogenous control.

### 4.10. Statistical Analysis

All data are shown as mean ± SEM. All statistical analyses were performed using GraphPad Prism v9 (San Diego, CA, USA). One-way ANOVA followed by Tukey’s multiple comparisons post hoc analysis was used to compare the data between the control, Ang II, and Ang II+Ang 1-7 groups; *p* < 0.05 was considered statistically significant.

## Figures and Tables

**Figure 1 ijms-23-15566-f001:**
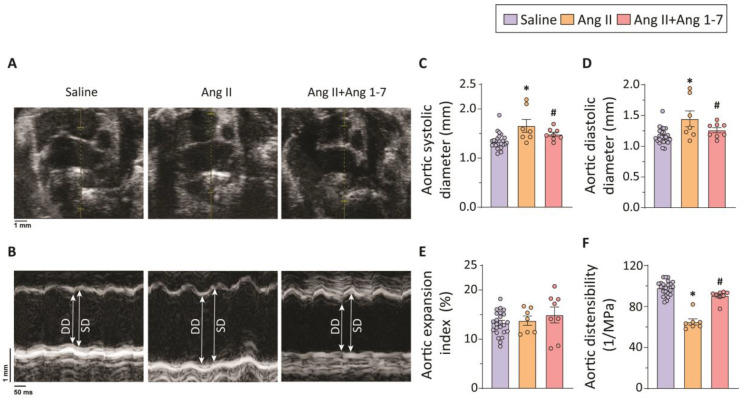
**ApoEKO mice developed TAA after four weeks of Angiotensin II (Ang II) infusion; chronic Ang 1-7 administration preserved structural and functional integrity of the thoracic aorta**. Representative B-mode (**A**) and M-mode (**B**) echocardiography images of the thoracic aorta and quantification of thoracic aortic systolic (**C**) and diastolic (**D**) diameters showing increased aortic dilatation in Ang II-infused ApoEKO mice (n = 7); chronic Ang 1-7 administration markedly reduced the aortic dilatation (n = 8) (**A**–**D**). Quantification of aortic expansion index (**E**) and aortic distensibility (**F**) exhibits unaltered aortic wall elasticity with markedly reduced distensibility in response to Ang II-infusion in ApoEKO mice; chronic Ang 1-7 administration reduced the aortic distensibility (**F**). * *p* < 0.05 compared with the saline group (n = 25); # *p* < 0.05 compared with the Ang II group using one-way ANOVA. DD: diastolic diameter, SD: systolic diameter.

**Figure 2 ijms-23-15566-f002:**
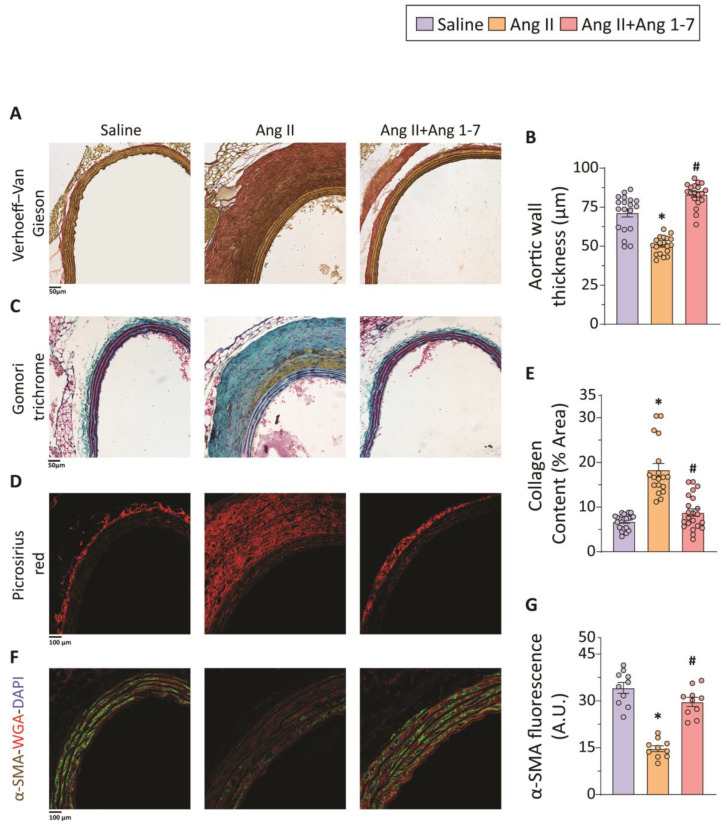
**Attenuation of Ang II-induced adverse thoracic aortic remodeling by Ang 1-7 infusion.** Representative images of Verhoeff–Van Gieson (elastin) staining and brightfield microscopy (**A**), and quantification of aortic media thickness (**B**) show significantly decreased medial thickness and increased medial degeneration in the thoracic aorta of ApoEKO mice in response to Ang II-infusion; chronic Ang 1-7 administration reduced medial degeneration and preserved thoracic aortic medial thickness. Representative images of Gomori trichrome staining and brightfield microscopy (**C**) and picrosirius red (collagen) staining and confocal microscopy (**D**), along with quantification of collagen content (**E**), indicate increased ECM remodeling in response to Ang II administration and abridged perivascular fibrosis resulting from chronic Ang 1-7 administration (**C**–**E**). Representative immunofluorescence and confocal microscopy images of α-SMA (green) and wheat-germ agglutinin (red) staining (**F**) of the thoracic aorta and quantification of α-SMA-immunoreactivity (**G**) exhibit reduced α-SMA fluorescence suggesting decreased contractile phenotype of VSMCs by Ang II administration and preservation of α-SMA levels by Ang 1-7 administration (**F**,**G**) in ApoEKO mice. Data points represent technical replicates from biological replicates (n = 3 in each group). * *p* < 0.05 compared with the saline group; # *p* < 0.05 compared with the Ang II group using one-way ANOVA.

**Figure 3 ijms-23-15566-f003:**
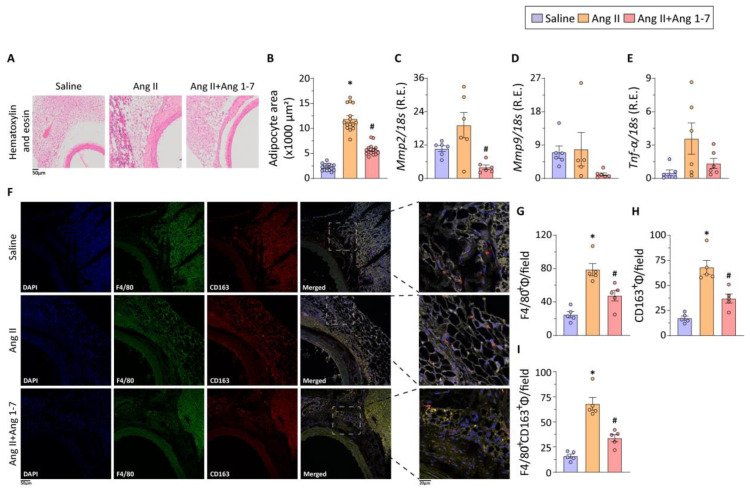
**Ang 1-7 reduces perivascular adipose tissue inflammation by attenuating macrophage maturation and polarization.** Representative images of hematoxylin and eosin (H&E) staining and brightfield microscopy (**A**) of the thoracic aorta and PVAT, along with quantification of adipocyte cross-sectional area (**B**), demonstrate increased adipocyte hypertrophy and inflammatory cell infiltration in periaortic adipose tissue in response to Ang II infusion; chronic Ang 1-7 administration reduced PVAT remodeling (**A**,**B**). TaqMan-qPCR-based mRNA expression analyses of matrix metalloproteinase 2 (*Mmp2*; (**C**)), matrix metalloproteinase 9 (*Mmp9*; (**D**)), and tumor necrosis factor-alpha (*Tnfa*; (**E**)) in perivascular adipose tissue show markedly increased expression of inflammatory genes in the Ang II-infusion group; Ang II-induced inflammation in thoracic PVAT is attenuated by Ang 1-7 administration (**C**–**E**). Representative images of F4/80 (green) and CD163 (red) double-immunostaining and confocal microscopy (**F**), along with their quantification (**F**–**H**), show increased infiltration of F4/80⁺(**G**), CD163⁺(**H**), and F4/80^+^CD163^+^ (**I**) macrophages, suggesting markedly increased inflammation in PVAT of Ang II-treated thoracic aorta; chronic administration of Ang 1-7 reduced F4/80^+^ and CD163^+^ macrophage in the peri-aortic adipose tissue (**F**–**I**). Data points represent technical replicates from biological replicates (n = 3 in each group). * *p* < 0.05 compared with the saline group; # *p* < 0.05 compared with the Ang II group using one-way ANOVA. Mϕ: macrophage.

**Figure 4 ijms-23-15566-f004:**
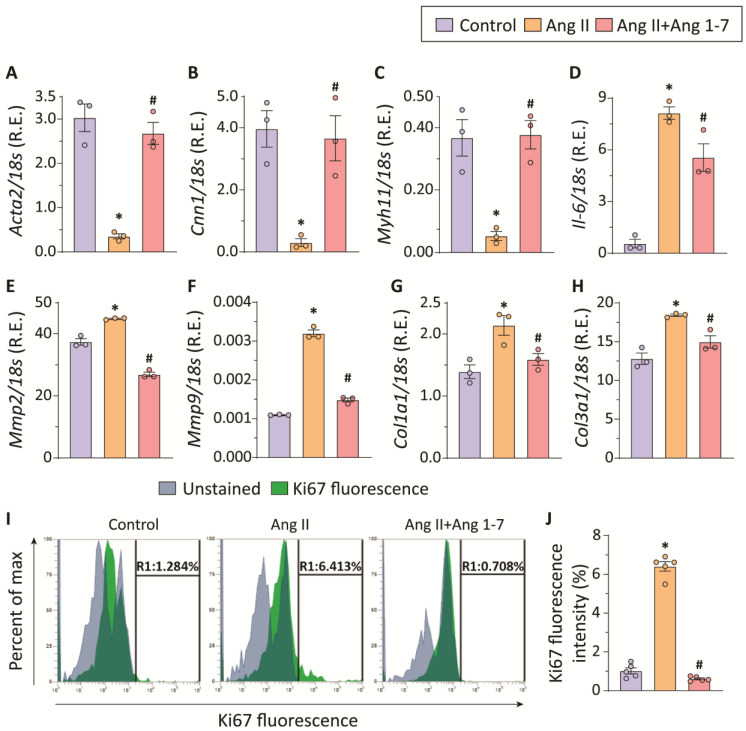
**Ang 1-7 attenuates Ang II-mediated phenotypic switching and hyperproliferation of thoracic aortic vascular smooth muscle cells (VSMCs).** TaqMan-qPCR-based mRNA expression analyses of alpha-smooth muscle actin (*Acta2*; (**A**)), calponin1 (*Cnn1*; (**B**)), myosin heavy chain 11 (*Myh11*; (**C**)), and interleukin-6 (*Il-6*; (**D**)) show the decreased expression of contractile genes and increased expression of the inflammatory gene in Ang II-treated thoracic SMCs. The mRNA expression analyses of matrix metalloproteinase 2 (*Mmp2*; (**E**)), matrix metalloproteinase 9 (*Mmp9*; (**F**)), collagen I (*Col1a1*; (**G**)), and collagen III (*Col3a1*; (**H**)) indicate pathological polarization of thoracic aortic SMCs to a “synthetic” phenotype in response to Ang II. Ang 1-7 treatment preserved the contractile phenotype of thoracic aortic SMCs (n = 3 in each group) (**A**–**H**). Flow cytometric analysis of Ki67 staining of thoracic aortic SMCs (**I**) and quantification of Ki67 fluorescence intensity (**J**) show increased Ki67-immunoreactivity in Ang II-treated VSMCs, indicative of hyperproliferation (n = 3 in each group). Chronic Ang 1-7 administration alleviated the Ang II-induced thoracic aortic SMC proliferation, as evident by decreased Ki67-immunoreactivity (**I**,**J**). * *p* < 0.05 compared with the control group; # *p* < 0.05 compared with the Ang II group using one-way ANOVA.

**Figure 5 ijms-23-15566-f005:**
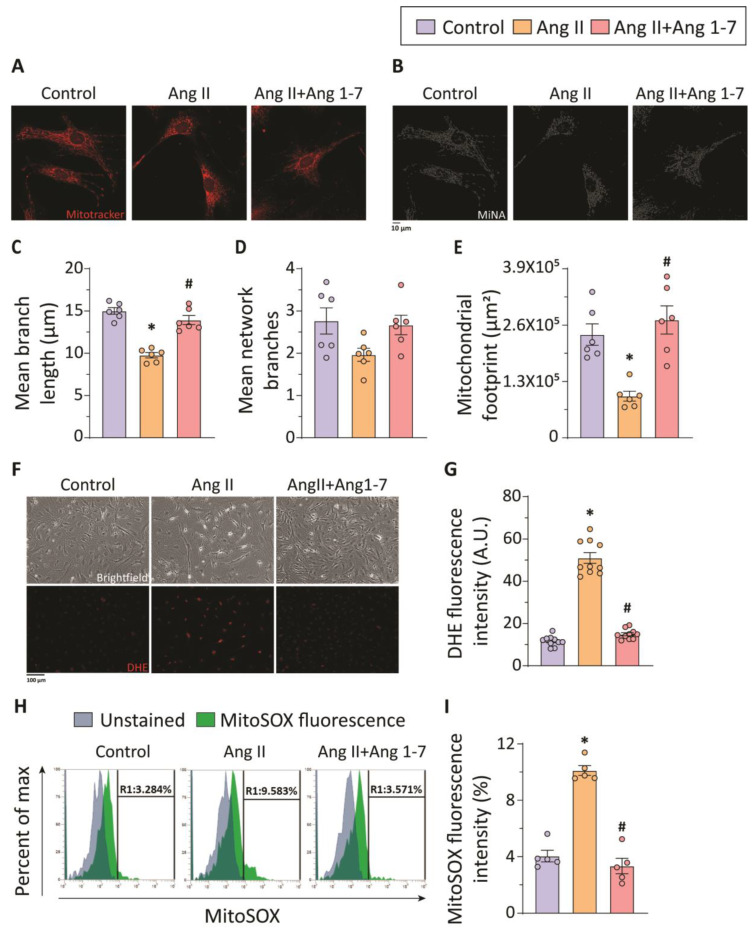
**Ang 1-7 attenuates Ang II-mediated mitochondrial fission and ROS generation in thoracic aortic SMCs.** Representative confocal images of Mitotracker red staining (**A**) and their analysis using the Mitochondrial Network Analysis program (MiNA plugin; ImageJ; (**B**)) show smaller and more fragmented mitochondria in response to Ang II-treated thoracic aortic SMCs compared to the control group (n = 6 in each group). Quantification of Mitotracker red-staining images using MiNA showing decreased mean branch length (**C**), mean network branches (**D**), and decreased mitochondrial footprint (**E**) in Ang II-treated VSMCs compared to control group. Ang 1-7 treatment attenuates Ang II-induced mitochondrial fission (**A**–**E**). Representative brightfield microscopy and confocal microscopy images of DHE-stained thoracic SMCs (**F**), along with quantification of DHE fluorescence intensity (**G**), show increased cellular ROS levels in Ang II-treated VSMCs compared to the control group (n = 5 in each group). Flow cytometric analysis of MitoSOX-stained thoracic aortic SMCs (**H**) and quantification of MitoSOX fluorescence (**I**) indicate increased mitochondrial ROS levels in Ang II-treated VSMCs (n = 5 in each group). Ang 1-7 treatment mitigates Ang II-mediated increased cellular and mitochondrial ROS levels in thoracic aortic SMCs (**F**–**I**). A.U.: arbitrary unit; * *p* < 0.05 compared with the control group; # *p* < 0.05 compared with Ang II group using one-way ANOVA.

**Figure 6 ijms-23-15566-f006:**
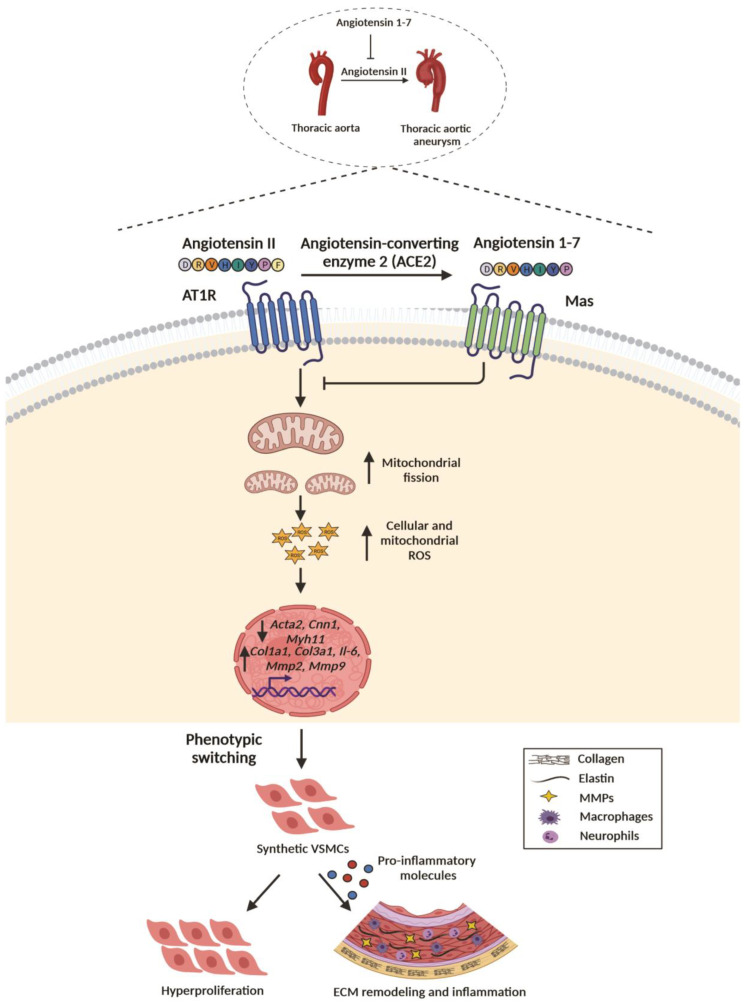
**Ang 1-7 attenuates TAA by antagonizing Ang II-mediated pathological changes in thoracic aortic SMCs.** Chronic infusion of Ang II, an effector protein of the renin-angiotensin system (RAS), leads to the development of thoracic aortic aneurysm (TAA) in ApoEKO mice. Mechanistically, Ang II treatment results in excessive mitochondrial fission in thoracic aortic SMCs, which consequently results in increased cellular and mitochondrial ROS. Ang II-induced oxidative stress affects the expression of various genes promoting the polarization of VSMCs to a synthetic, proliferative, and inflammatory phenotype. Ang 1-7 antagonizes pathogenic effects of Ang II and attenuates excessive mitochondrial fission, oxidative stress, inflammation, and phenotypic switching of thoracic aortic SMCs, resulting in attenuation of TAA development in ApoEKO mice. Figure created using Biorender.com.

## Data Availability

The data that support the findings of this study are available from the corresponding author upon reasonable request.

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
