# Peer review of "Attenuation of Smooth Muscle Cell Phenotypic Switching by Angiotensin 1-7 Protects against Thoracic Aortic Aneurysm"

_ijms, 2022, doi:10.3390/ijms232415566_

Round 1

Reviewer 1 Report

This is an interesting article dealing with phenotypic switching of VSMCs isolated from the aorta of murine model of TAA. The study reports the significant therapeutic effects of the heptapeptide Ang 1-7 in all the assays performed. It is rather surprising to read how well it works simply missing single amino acid from the octapeptide AngII. The working model is suitable to carry out this study. 

Introduction.- Some information is missing of previous works about phenotypic switching in VSMCs in TAA. For instance, in Marfan syndrome, which is considered a TAA, and it is not considered at all. Other point is the fact that they say that there is none current clinically approved therapy from TAA. This is not the case again in the case of Marfan syndrome, which as said above are also thoracic aortopathies.

Material and Methods. Results are shown as mean +/- SEM, but statistically speaking, it is more correct to express this type of results as the mean +/- SD. Is there any reason for not showing SD but SEM?

Results.- They are well designed, presented and explained. I suggest to include some seminal references when they affirm that "VSMC phenotypic switch has been consistently observed to contribute to the development and progression of various diseases, including aortic aneurysms". It would be of great interest to know how Ang1-7 treatment affects the canonical and non-canonical TGF-beta signaling in the isolated aortic VSMCs and if they are associated to the phenotypic switch in untreated and Ang1-7 treated animals.

Discussion.- It is well structured whereas I miss more explanations about what the molecular mechanisms might be linking some observations. For instance, the fission of mitochondria with the phenotypic switching and ROS generation or its connection with lipotoxicity.

Figures.- I warmly suggest enlarging some picture with immunofluorescence to easily visualize the results. For instance, Figs. 5A and B. 

Author Response

Response to Editor and Reviewers

Reviewer 1:

This is an interesting article dealing with phenotypic switching of VSMCs isolated from the aorta of murine model of TAA. The study reports the significant therapeutic effects of the heptapeptide Ang 1-7 in all the assays performed. It is rather surprising to read how well it works simply missing single amino acid from the octapeptide AngII. The working model is suitable to carry out this study. 

Response: We thank the reviewer for his/her suggestions and constructive feedback. 

Introduction: Some information is missing of previous works about phenotypic switching in VSMCs in TAA. For instance, in Marfan syndrome, which is considered a TAA, and it is not considered at all. Other point is the fact that they say that there is none current clinically approved therapy from TAA. This is not the case again in the case of Marfan syndrome, which as said above are also thoracic aortopathies.

Response: We thank the reviewer for the constructive feedback. As our manuscript primarily focuses on the pathogenic contributions of vascular smooth muscle cell phenotypic switching, in absence of hereditary syndromes such as Marfan Syndrome, Ehlers-Danlos Syndrome, Loeys-Dietz Syndrome, and Turner Syndrome (Line 425-427), we have not discussed TAA pathophysiology in these genetic disorders in the manuscript. Medications such as beta-blockers, Angiotensin II receptor blockers, and statins have been prescribed for the management of TAA, but as surgery remains the only viable alternative in the severe form of TAA, our focus lies on identification of novel treatment which can attenuate the onset and progression of TAA by limiting pathogenic alterations in the thoracic aorta.

We have added the following to our revised discussion.

“The VSMC phenotypic switching has been identified to be a key mechanistic observation in various aortopathies, including Marfan Syndrome, Ehlers-Danlos Syndrome, Loeys-Dietz Syndrome, and Turner Syndrome (1-4).” Please see page 19 of the revised manuscript.

Material and Methods: Results are shown as mean +/- SEM, but statistically speaking, it is more correct to express this type of results as the mean +/- SD. Is there any reason for not showing SD but SEM?

Response: We would like to thank the reviewer for the feedback. We agree with the reviewer that though SEM and SD are used interchangeably, SD and SEM estimate data differently. SD is an index of variability in the data which may be deceptive in an experimental situation where high variability in the biological data differs from the normal data distribution. Thus, we have depicted individual data points representing replicates with SEM to indicate the precision of the study, and how well the biological replicates truly represent the entire population (5). In our opinion, representation of individual data points presents information in much more transparent manner, which can not be achieved by only incorporating Mean and SD (or SEM).

Results: They are well designed, presented and explained. I suggest including some seminal references when they affirm that "VSMC phenotypic switch has been consistently observed to contribute to the development and progression of various diseases, including aortic aneurysms". It would be of great interest to know how Ang1-7 treatment affects the canonical and non-canonical TGF-beta signaling in the isolated aortic VSMCs and if they are associated to the phenotypic switch in untreated and Ang1-7 treated animals.

Response: To discuss the causal association of VSMCs phenotypic switching with vascular inflammation and disease onset and progression, we have included relevant references in the revised results section. The appended references are highlighted in the tracked change manuscript and the bibliography is accordingly amended. We agree with the reviewer that it will be exciting to study the protective effects of Ang 1-7 via alteration in canonical and non-canonical TGF-β pathway in the context of TAA as TGF-β signaling represents one of the major pathogenic pathways in TAA development. Being a multifactorial disease, investigation of other mechanisms of TAA development requires further studies. Our future experiments will aim to investigate the protective role of Ang 1-7 in suggested pathways.  

We have added the following to our revised results.

Synthetic VSMCs promote ECM remodeling and vascular inflammation, playing a pivotal role in the growth of an aortic aneurysm (6-8).” Please see page 15 of the revised manuscript.

Discussion: It is well structured whereas I miss more explanations about what the molecular mechanisms might be linking some observations. For instance, the fission of mitochondria with the phenotypic switching and ROS generation or its connection with lipotoxicity.

Response: We would like to thank the reviewer for the feedback. We have provided relevant references in the manuscript linking mitochondrial fission and ROS generation with VSMCs phenotypic switching.

We have added the following to our revised manuscript.

“The study emphasized the role of Drp1 phosphorylation in increased mitochondrial fission leading to Ang II-stimulated VSMC phenotypic switching. Similarly, Lu et.al. demonstrated Drp1-dependent Ang II-induced VSMC phenotypic switching and attenuation by diallyl trisulfide  (9).”  Please see page 20 of the revised manuscript.

“Accumulating evidence suggested an association of the generation of ROS with aberrant mitochondrial morphology (10). Conversely, genetic ablation of pro-mitofusion genes or inducible deletion of Drp1 leads to altered mitochondrial morphology and generation of ROS (11-13). Mechanistically, we speculate a reciprocal association between mitochondrial dynamics and ROS generation during Ang II-induced VSMCs phenotypic switching in TAA.” Please see pages 20-21 of the revised manuscript.

“The Activation of the Ang 1-7/Mas axis activation leads to reduced expression of proinflammatory cytokines. The multifactorial protective effects of Ang 1-7 treatment are attributed to the amelioration of PVAT inflammation, activation, and polarization of macrophages.” Please see page 21 of the revised manuscript.

Figures: I warmly suggest enlarging some picture with immunofluorescence to easily visualize the results. For instance, Figs. 5A and B. 

Response: We would like to thank the reviewer for the suggestion. We have acquired the immunofluorescence images (Figures 5A and B) at 63X magnification on the SP8 confocal microscope and provided high resolution (300 dpi) images for the publication. We have purposely kept these configurations as further magnification will not allow capturing the complete vascular smooth muscle cell with mitochondrial changes due to a lesser field of view.

Reviewer 2:

Jadli et al., examined the impact of Ang 1-7 treatment on AngII promoted vascular remodeling and TAA and report Ang 1-7 to be protective by mitigating VSMC (and PVAT) inflammation, fibrotic responses, ROS production, and cell proliferation. Several comments should be addressed to validate the proposed Ang 1-7 effects and to underpin the conclusions drawn.  

Response: We thank the reviewer for the constructive feedback and suggestions to improve the submitted manuscript. We have tried to answer all the queries raised by the reviewer and made the necessary changes in the revised manuscript.

Minor comments:

  1. Could the authors explain the pretty varying n-number amongst the groups in the echo data? While the control group consists of n around 20, treatment groups are represented by n=7-8. Echo analyses of all mice examined for other parameters would have offered the possibility to also correlate dilation severity with molecular and histological findings.

Response: We thank the reviewer for the constructive feedback. We constantly strive to follow the recommendations by Canadian Council on Animal Care (CCAC) for animal usage on our studies. Particularly, Three Rs principle recommends actively looking for ways to get find approaches which would allow “reduction, replacement, and refinement” in animal studies. Accordingly, in addition to phenotyping studies, the mice from control groups were also used for VSMCs isolation for in-vitro experiments after completing the echo analyses. The mice from Ang II and Ang II-Ang 1-7 groups were dedicated to in vivo and ex vivo experiments. We have presented the echo data from all the mice included in the present study, which were then subdivided to various molecular and histological analyses.

  1. AngII was subcutaneously infused while Ang 1-7 was administered via injections. Wouldn’t it be possible to combine the two peptides in one pump?

Response: As mentioned in the methods section, while Ang II pumps were implanted subcutaneously, Ang 1-7 pumps were implanted intraperitoneally. Due to the concentration used and filling capacity of the available osmotic pumps and accepted scientific practice associated with Ang II and Ang 1-7 osmotic pumps, we have not combined these two peptides in a single pump during surgery. Moreover, the combination of both peptides in a single pump may have some unfounded and unknown interactions, which has a potential to resulting in altered release rate or activity, leading to confounding effects.

  1. In Fig. 2A+C, the images representing the AngII+Ang 1-7 group seem somehow magnified. The authors may consider replacing the respective images and/or adding the scale bar.

Response: We have now replaced images (Fig. 2A & C) and amended the figure file with the updated images.

  1. Sustained infusion with 1.44 mg/kg/day AngII does not only induce TAA but also mediates hemodynamic alterations and concentric LV hypertrophy - what is known about the interference of Ang 1-7 with the afterload triggered remodeling process?

Response: We thank the reviewer for the constructive feedback. The reviewer has correctly mentioned the detrimental role of chronic administration of Ang II in hemodynamic alterations and the development of cardiac hypertrophy. Though it was out of scope for the current manuscript, we have previously shown antihypertrophic and cardioprotective properties of Ang (1–7) by attenuation of Ang II-induced cardiac hypertrophy (14-16). Chronic administration of Ang 1-7 has shown to attenuate Ang II-induced cardiac hypertrophy and fibrosis by a mitochondrial ROS-dependent mechanism (15, 17). Thus, Ang 1-7 is reported to have cardioprotective effects against Ang II-induced cardiac remodeling.

Major comments:

  1. To assess the impact of Ang 1-7 on AngII-mediated changes in mitochondrial structure, ROS formation, and increased cell proliferation, in vitro experiments on VSMCs isolated from naïve mice were performed. Isolation and analyses of VSMCs or tissue immunohistochemistry (for DHE staining) from mice upon sustained AngII infusion +/- Ang 1-7 injection would be crucial to determine the impact of Ang 1-7 on the cellular changes associated with TAA formation and progression.

Response: We would like to thank the reviewer for the constructive feedback and suggestion. We have used Ang II-infusion murine model for TAA and in vitro effects of Ang 1-7 on Ang II-mediated pathogenic changes were studied using VSMCs isolated from ApoEKO mice. Though we have not isolated VSMCs from Ang II and Ang II+Ang 1-7 infusion groups, the isolated cells were treated to mimic in vivo conditions by 24 hr Ang II and Ang II+Ang 1-7 treatment. The in vitro experiments were performed in accordance with previously published studies (18, 19).

  1. The text referring to Fig. 3C-E states a significant increase in Mmp-2, -9, Tnfa  PVAT mRNA expression and the prevention thereof by Ang 1-7, yet the statistical analysis does not support this statement. In the respective paragraph, UCP-1 expression is mentioned, and that TAA leads to adipose tissue whitening – when emphasizing this regulation, why not measuring UCP-1 mRNA expression?

Response: We have amended the text indicating that though there is decreased mRNA expression of Mmp2, Mmp9, and Tnfa with Ang 1-7 treatment, results did not reach statistical significance. The browning of PVAT and expression of brown adipocyte marker UCP-1 expression were included to define characteristics of thoracic PVAT. As our study was focused on Ang II-mediated inflammation in PVAT and its attenuation by Ang 1-7, we did not investigate other properties of PVAT. As the suggested experiment may provide insight into other pathways of TAA onset, we will aim to investigate these alterations in future studies. Thank you.

We have added the following to our revised results.

“Though, mRNA expression for Mmp2, Mmp9, and Tnfa were not statistically significant, Ang 1-7-mediated alleviation of adipose tissue remodeling also resulted in decreased mRNA expressions of Mmp2, Mmp9, and Tnfa (Figure 3C-E)." Please see page 14 of the revised manuscript.  

  1. F4/80 does not represent a marker for resident macrophages but rather a pan macrophage marker – the respective sentence should be rephrased. In Fig. 3F cells positive for the different markers are barely visible due to the strong background/autofluorescence signal in the different channels. To quantify macrophage-subtypes a quantification of double-positive cells would be suggested (F4/80+ CD163+).  Moreover, while AngII clearly increases macrophage counts in general, the assumption that AngII also promotes (and Ang 1-7 prevents) their polarization toward a CD163+ phenotype is not supported by the results shown. Not only in the treatment groups but also in the control group 80-90% of all macrophages (F4/80+) exhibit positivity for CD163.

Response: We have rephrased the respective sentence in the revised manuscript. We have calculated the F4/80+CD163+ dual positive macrophages in the thoracic aorta (revised Figure 3I). Our study demonstrated attenuation of macrophage infiltration and polarization by Ang 1-7 treatment. Though lowest compared to other groups, the control group also showed macrophages and their polarization in the thoracic aorta, the observation is consistent with previous studies indicating baseline activity of macrophages (20, 21).

We have added the following to our revised results.

“Immunofluorescence staining and confocal imaging of the PVAT displayed markedly increased levels of pan macrophage marker F4/80+ in response to Ang II administration (Figure 3F-G).” Please see page 14 of the revised manuscript.

  1. The authors state an anti-inflammatory action of Ang 1-7; however, this conclusion is merely based on Il-6 mRNA expression on which Ang 1-7 shows a far milder effect than on all other parameters assessed. Moreover, most presumably Il-6 mRNA expression is significantly higher in the AngII+Ang 1-7 group compared to the control group which should be indicated. To emphasize an anti-inflammatory role of Ang 1-7 flow cytometry analyses of excised aortas would be recommended.

Response: We agree with the reviewer's comment on Ang 1-7 mediated attenuation of Il-6 and have made changes in the manuscript to clarify.

We have added the following to the revised results.

" Though Ang 1-7 treatment attenuated Ang II-induced elevated Il-6 expression, the higher mRNA expression of Il-6 in Ang 1-7 group compared to controls suggested milder effect of Ang 1-7 on inflammatory cytokines than other parameters.  (Figure 4D-H).” Please see page 15 of the revised manuscript.

“Thus, gene expression data and attenuation of Ang II-induced macrophage infiltration in the thoracic aorta suggested anti-inflammatory effects of Ang 1-7 in TAA.” Please see page 16 of the revised manuscript.

  1. In the discussion section the authors indicate that numerous studies have identified Ang 1-7 effects to antagonize AngII-induced detrimental effects in cardiovascular pathologies. The referenced original articles report cerebrovascular effects of Ang 1-7, expression-regulation of Ang 1-7 in Nrf2 KO mice, and the effect of an Ang 1-7 antagonist on atherosclerotic plaques. More adequate references should be chosen to support the statement.

Response: We have added relevant references to reinforce the statement that “numerous studies have identified Ang 1-7 effects to antagonize Ang II-induced detrimental effects in cardiovascular pathologies” in the discussion section.

We have added the following to the revised discussion.

“Numerous studies have identified Ang 1-7 effects that antagonize Ang II-induced detrimental defects in cardiovascular pathologies (14, 15, 22-27).” Please see page 18 of the revised manuscript.

References:

  1. Mortensen KH, Andersen NH, Gravholt CH. Cardiovascular phenotype in Turner syndrome--integrating cardiology, genetics, and endocrinology. Endocrine reviews. 2012;33(5):677-714.
  2. Dale M, Fitzgerald MP, Liu Z, Meisinger T, Karpisek A, Purcell LN, et al. Premature aortic smooth muscle cell differentiation contributes to matrix dysregulation in Marfan Syndrome. PloS one. 2017;12(10):e0186603.
  3. Fletcher AJ, Syed MBJ, Aitman TJ, Newby DE, Walker NL. Inherited Thoracic Aortic Disease: New Insights and Translational Targets. Circulation. 2020;141(19):1570-87.
  4. Crosas-Molist E, Meirelles T, López-Luque J, Serra-Peinado C, Selva J, Caja L, et al. Vascular smooth muscle cell phenotypic changes in patients with Marfan syndrome. Arteriosclerosis, thrombosis, and vascular biology. 2015;35(4):960-72.
  5. Barde MP, Barde PJ. What to use to express the variability of data: Standard deviation or standard error of mean? Perspectives in clinical research. 2012;3(3):113-6.
  6. Alexander MR, Owens GK. Epigenetic control of smooth muscle cell differentiation and phenotypic switching in vascular development and disease. Annual review of physiology. 2012;74:13-40.
  7. Milewicz DM, Trybus KM, Guo DC, Sweeney HL, Regalado E, Kamm K, et al. Altered Smooth Muscle Cell Force Generation as a Driver of Thoracic Aortic Aneurysms and Dissections. Arteriosclerosis, thrombosis, and vascular biology. 2017;37(1):26-34.
  8. Malashicheva A, Kostina D, Kostina A, Irtyuga O, Voronkina I, Smagina L, et al. Phenotypic and Functional Changes of Endothelial and Smooth Muscle Cells in Thoracic Aortic Aneurysms. International journal of vascular medicine. 2016;2016:3107879.
  9. Lu ZY, Qi J, Yang B, Cao HL, Wang RY, Wang X, et al. Diallyl Trisulfide Suppresses Angiotensin II-Induced Vascular Remodeling Via Inhibition of Mitochondrial Fission. Cardiovascular drugs and therapy. 2020;34(5):605-18.
  10. Willems PH, Rossignol R, Dieteren CE, Murphy MP, Koopman WJ. Redox Homeostasis and Mitochondrial Dynamics. Cell metabolism. 2015;22(2):207-18.
  11. Muñoz JP, Ivanova S, Sánchez-Wandelmer J, Martínez-Cristóbal P, Noguera E, Sancho A, et al. Mfn2 modulates the UPR and mitochondrial function via repression of PERK. The EMBO journal. 2013;32(17):2348-61.
  12. Tang S, Le PK, Tse S, Wallace DC, Huang T. Heterozygous mutation of Opa1 in Drosophila shortens lifespan mediated through increased reactive oxygen species production. PloS one. 2009;4(2):e4492.
  13. Santoro A, Campolo M, Liu C, Sesaki H, Meli R, Liu ZW, et al. DRP1 Suppresses Leptin and Glucose Sensing of POMC Neurons. Cell metabolism. 2017;25(3):647-60.
  14. Patel VB, Bodiga S, Fan D, Das SK, Wang Z, Wang W, et al. Cardioprotective effects mediated by angiotensin II type 1 receptor blockade and enhancing angiotensin 1-7 in experimental heart failure in angiotensin-converting enzyme 2-null mice. Hypertension (Dallas, Tex : 1979). 2012;59(6):1195-203.
  15. Guo L, Yin A, Zhang Q, Zhong T, O'Rourke ST, Sun C. Angiotensin-(1-7) attenuates angiotensin II-induced cardiac hypertrophy via a Sirt3-dependent mechanism. American journal of physiology Heart and circulatory physiology. 2017;312(5):H980-h91.
  16. McCollum LT, Gallagher PE, Ann Tallant E. Angiotensin-(1-7) attenuates angiotensin II-induced cardiac remodeling associated with upregulation of dual-specificity phosphatase 1. American journal of physiology Heart and circulatory physiology. 2012;302(3):H801-10.
  17. Altobelli E, Rapacchietta L, Profeta VF, Fagnano R. Risk Factors for Abdominal Aortic Aneurysm in Population-Based Studies: A Systematic Review and Meta-Analysis. Int J Environ Res Public Health. 2018;15(12).
  18. Patel VB, Zhong JC, Fan D, Basu R, Morton JS, Parajuli N, et al. Angiotensin-converting enzyme 2 is a critical determinant of angiotensin II-induced loss of vascular smooth muscle cells and adverse vascular remodeling. Hypertension (Dallas, Tex : 1979). 2014;64(1):157-64.
  19. Xue F, Yang J, Cheng J, Sui W, Cheng C, Li H, et al. Angiotensin-(1-7) mitigated angiotensin II-induced abdominal aortic aneurysms in apolipoprotein E-knockout mice. British journal of pharmacology. 2020;177(8):1719-34.
  20. Moore JP, Vinh A, Tuck KL, Sakkal S, Krishnan SM, Chan CT, et al. M2 macrophage accumulation in the aortic wall during angiotensin II infusion in mice is associated with fibrosis, elastin loss, and elevated blood pressure. American journal of physiology Heart and circulatory physiology. 2015;309(5):H906-17.
  21. Dale MA, Xiong W, Carson JS, Suh MK, Karpisek AD, Meisinger TM, et al. Elastin-Derived Peptides Promote Abdominal Aortic Aneurysm Formation by Modulating M1/M2 Macrophage Polarization. Journal of immunology (Baltimore, Md : 1950). 2016;196(11):4536-43.
  22. Zhao S, Ghosh A, Lo CS, Chenier I, Scholey JW, Filep JG, et al. Nrf2 Deficiency Upregulates Intrarenal Angiotensin-Converting Enzyme-2 and Angiotensin 1-7 Receptor Expression and Attenuates Hypertension and Nephropathy in Diabetic Mice. Endocrinology. 2018;159(2):836-52.
  23. Patel VB, Zhong JC, Grant MB, Oudit GY. Role of the ACE2/Angiotensin 1-7 Axis of the Renin-Angiotensin System in Heart Failure. Circulation research. 2016;118(8):1313-26.
  24. Zhang YH, Zhang YH, Dong XF, Hao QQ, Zhou XM, Yu QT, et al. ACE2 and Ang-(1-7) protect endothelial cell function and prevent early atherosclerosis by inhibiting inflammatory response. Inflammation research : official journal of the European Histamine Research Society [et al]. 2015;64(3-4):253-60.
  25. Silva GM, França-Falcão MS, Calzerra NTM, Luz MS, Gadelha DDA, Balarini CM, et al. Role of Renin-Angiotensin System Components in Atherosclerosis: Focus on Ang-II, ACE2, and Ang-1-7. Frontiers in physiology. 2020;11:1067.
  26. Peña Silva RA, Kung DK, Mitchell IJ, Alenina N, Bader M, Santos RA, et al. Angiotensin 1-7 reduces mortality and rupture of intracranial aneurysms in mice. Hypertension (Dallas, Tex : 1979). 2014;64(2):362-8.
  27. Grobe JL, Mecca AP, Lingis M, Shenoy V, Bolton TA, Machado JM, et al. Prevention of angiotensin II-induced cardiac remodeling by angiotensin-(1-7). American journal of physiology Heart and circulatory physiology. 2007;292(2):H736-42.

Reviewer 2 Report

Jadli et al., examined the impact of Ang 1-7 treatment on AngII promoted vascular remodelling and TAA and report Ang 1-7 to be protective by mitigating VSMC (and PVAT) inflammation, fibrotic responses, ROS production and cell proliferation. Several comments should be addressed to validate the proposed Ang 1-7 effects and to underpin the conclusions drawn.  

Minor comments:

1.       Could the authors explain the pretty varying n-number amongst the groups in the echo data? While control group consists of n around 20, treatment groups are represented by n=7-8. Echo analyses of all mice examined for other parameters would have offered the possibility to also correlate dilation severity with molecular and histological findings.

2.       AngII was subcutaneously infused while Ang 1-7 was administered via injections. Wouldn’t it be possible to combine the two peptides in one pump?

3.       In Fig. 2A+C, the images representing the AngII+Ang 1-7 group seem somehow magnified. The authors may consider replacing the respective images and/or adding the scale bar.

4.       Sustained infusion with 1.44 mg/kg/day AngII does not only induce TAA but also mediates hemodynamic alterations and concentric LV hypertrophy - what is known about the interference of Ang 1-7 with the after-load triggered remodelling process?

Major comments:

1.       To assess the impact of Ang 1-7 on AngII mediated changes in mitochondrial structure, ROS formation, and increased cell proliferation, in vitro experiments on VSMCs isolated from naïve mice were performed. Isolation and analyses of VSMCs or tissue immunohistochemistry (for DHE staining) from mice upon sustained AngII infusion +/- Ang 1-7 injection would be crucial to determine the impact of Ang 1-7 on the cellular changes associated with TAA formation and progression.

2.       The text referring to Fig. 3C-E states a significant increase in Mmp-2, -9, Tnfa  PVAT mRNA expression and the prevention thereof by Ang 1-7, yet the statistical analysis does not support this statement. In the respective paragraph, UCP-1 expression is mentioned, and that TAA leads to adipose tissue whitening – when emphasizing this regulation, why not measuring UCP-1 mRNA expression?

3.       F4/80 does not represent a marker for resident macrophages but rather a pan macrophage marker – the respective sentence should be rephrased. In Fig. 3F cells positive for the different markers are barely visible due to the strong background/autofluorescence signal in the different channels. To quantify macrophage-subtypes a quantification of double-positive cells would be suggested (F4/80+ CD163+).  Moreover, while AngII clearly increases macrophage counts in general, the assumption that AngII also promotes (and Ang 1-7 prevents) their polarization toward a CD163+ phenotype is not supported by the results shown. Not only in the treatment groups but also in the control group 80-90% of all macrophages (F4/80+) exhibit positivity for CD163.

4.       The authors state an anti-inflammatory action of Ang 1-7, however, this conclusion is merely based on Il-6 mRNA expression on which Ang 1-7 shows a far milder effect than on all other parameters assessed. Moreover, most presumably Il-6 mRNA expression is significantly higher in the AngII+Ang 1-7 group compared to the control group which should be indicated. To emphasize an anti-inflammatory role of Ang 1-7 flow cytometry analyses of excised aortas would be recommended.

5.       In the discussion section the authors indicate that numerous studies have identified Ang 1-7 effects to antagonize AngII-induced detrimental effects in cardiovascular pathologies. The referenced original articles report cerebrovascular effects of Ang 1-7, expression-regulation of Ang 1-7 in Nrf2 KO mice, and the effect of an Ang 1-7 antagonist on atherosclerotic plaques. More adequate references should be chosen to support the statement.

Author Response

(The authors gave the same response as above.)

Round 2

Reviewer 2 Report

The authors have addressed all comments raised by replacing representative images, recalculating macrophage numbers as F4/80+CD163+ double-positive cells and by re-phrasing respective sentences in the manuscript. However, two of the initially raised comments shall be addressed more adequately:

2. The text referring to Fig. 3C-E states a significant increase in Mmp-2, -9, Tnfa PVAT mRNA

expression and the prevention thereof by Ang 1-7, yet the statistical analysis does not support this

statement. In the respective paragraph, UCP-1 expression is mentioned, and that TAA leads to

adipose tissue whitening – when emphasizing this regulation, why not measuring UCP-1 mRNA

expression?

Response: We have amended the text indicating that though there is decreased mRNA expression

of Mmp2, Mmp9, and Tnfa with Ang 1-7 treatment, results did not reach statistical significance.

The browning of PVAT and expression of brown adipocyte marker UCP-1 expression were

included to define characteristics of thoracic PVAT. As our study was focused on Ang II-mediated

inflammation in PVAT and its attenuation by Ang 1-7, we did not investigate other properties of

PVAT. As the suggested experiment may provide insight into other pathways of TAA onset, we

will aim to investigate these alterations in future studies. Thank you.

We have added the following to our revised results.

“Though, mRNA expression for Mmp2, Mmp9, and Tnfa were not statistically significant, Ang 1-

7-mediated alleviation of adipose tissue remodeling also resulted in decreased mRNA expressions

of Mmp2, Mmp9, and Tnfa (Figure 3C-E)." Please see page 14 of the revised manuscript.

Currently, an Ang1-7 mediated decrease in mRNA expression is only true for MMP-2 while MMP-9 and Tnfa are most presumably not even showing a statistical trend toward a regulation - this should be reflected by the interpretation of the data. Would the authors have the chance to increase n-numbers for the qPCR to reduce SEM and thereby verify the AngII mediated up-regulation and the Ang1-7 mediated protection? According to echo data, the treatment groups should encompass at least one more individual per group.

Would there still be tissue or cDNA available to actually run the PCR for the emphasized UCP-1 marker and assess whether Ang1-7 might also positively impact on it?

3. F4/80 does not represent a marker for resident macrophages but rather a pan macrophage marker

– the respective sentence should be rephrased. In Fig. 3F cells positive for the different markers

are barely visible due to the strong background/autofluorescence signal in the different channels.

To quantify macrophage-subtypes a quantification of double-positive cells would be suggested

(F4/80+ CD163+). Moreover, while AngII clearly increases macrophage counts in general, the

assumption that AngII also promotes (and Ang 1-7 prevents) their polarization toward a CD163+

phenotype is not supported by the results shown. Not only in the treatment groups but also in the

control group 80-90% of all macrophages (F4/80+) exhibit positivity for CD163.

Response: We have rephrased the respective sentence in the revised manuscript. We have

calculated the F4/80+CD163+ dual positive macrophages in the thoracic aorta (revised Figure

3I). Our study demonstrated attenuation of macrophage infiltration and polarization by Ang 1-7

treatment. Though lowest compared to other groups, the control group also showed macrophages

and their polarization in the thoracic aorta, the observation is consistent with previous studies

indicating baseline activity of macrophages (20, 21).

We have added the following to our revised results.

“Immunofluorescence staining and confocal imaging of the PVAT displayed markedly increased

levels of pan macrophage marker F4/80+ in response to Ang II administration (Figure 3F-G).”

Please see page 14 of the revised manuscript.

CD163+ macrophages are stated to be strongly associated with a fibrotic (M2) response, yet, their up-regulation concluded to reflect inflammation - which is a bit misleading. Could the authors please revise the respective section?

Author Response

Please see the attached file for our detailed response to reviewer 2 comments.
